# HbA1c recording in patients following a first diagnosis of serious mental illness: the South London and Maudsley Biomedical Research Centre case register

Nikeysha Bell ,[1] Gayan Perera ,[1] David Chandran ,[1] Brendon Stubbs ,[2,3] Fiona Gaughran ,[2,3] Robert Stewart [1,3]

¹Psychological Medicine, King's College London Institute of Psychiatry, Psychology and Neuroscience, London, UK
²Institute of Psychiatry, Psychology and Neuroscience, Psychosis Studies, King's College London, London, UK
³National Psychosis Service, South London and Maudsley NHS Foundation Trust, London, UK

**Correspondence to**
Dr Nikeysha Bell;
nikeysha.bell@kcl.ac.uk

## ABSTRACT

**Objectives** To investigate factors associated with the recording of glycated haemoglobin (HbA1c) in people with first diagnoses of serious mental illness (SMI) in a large mental healthcare provider, and factors associated with HbA1c levels, when recorded. To our knowledge this is the first such investigation, although attention to dysglycaemia in SMI is an increasing priority in mental healthcare.

**Design** The study was primarily descriptive in nature, seeking to ascertain the frequency of HbA1c recording in the mental healthcare sector for people following first SMI diagnosis.

**Settings** A large mental healthcare provider, the South London and Maudsley National Health Service Trust.

**Participants** Using electronic mental health records data, we ascertained patients with first SMI diagnoses (schizophrenia, schizoaffective disorder, bipolar disorder) from 2008 to 2018.

**Outcome measures** Recording or not of HbA1c level was ascertained from routine local laboratory data and supplemented by a natural language processing (NLP) algorithm for extracting recorded values in text fields (precision 0.89%, recall 0.93%). Age, gender, ethnic group, year of diagnosis, and SMI diagnosis were investigated as covariates in relation to recording or not of HbA1c and first recorded levels.

**Results** Of 21 462 patients in the sample (6546 bipolar disorder; 14 916 schizophrenia or schizoaffective disorder; mean age 38.8 years, 49% female), 4106 (19.1%) had at least one HbA1c result recorded from laboratory data, increasing to 6901 (32.2%) following NLP. HbA1c recording was independently more likely in non-white ethnic groups (black compared with white: OR 2.45, 95% CI 2.29 to 2.62), and was negatively associated with age (OR per year increase 0.93, 0.92–0.95), female gender (0.83, 0.78–0.88) and bipolar disorder (0.49, 0.45–0.52).

**Conclusions** Over a 10-year period, relatively low level of recording of HbA1c was observed, although this has increased over time and ascertainment was increased with text extraction. It remains important to improve the routine monitoring of dysglycaemia in these at-risk disorders.

---

**STRENGTHS AND LIMITATIONS OF THIS STUDY**

⇒ We believe this is the first study of this kind.
⇒ Seeking to understand whether HbA1c is being recorded/sufficiently recorded in the serious mental illness population (a current National Health Service target).
⇒ High precision and recall for the natural language processing algorithm (precision 0.89%, recall 0.93%).
⇒ Focus on specific catchment area.
⇒ Does not account for information within primary care.

---

## BACKGROUND

People with serious mental illness (SMI) have a substantially reduced life expectancy[1] presenting a major public health challenge. Among a range of specific health inequalities, type 2 diabetes mellitus (T2DM) incidence rates are higher than expected,[2] potentially accounting for at least some of the mortality gap and providing an important opportunity for health improvement initiatives. As well as unfavourable lifestyle factors, antipsychotic medications may play at least some part in the high prevalence of T2DM through accelerating glucose dysregulation.[3] Other factors such as diagnostic overshadowing and under-screening and subsequent treatment could also compound this issue and have been demonstrated in people with established SMI with cardiovascular disease.[4] Identifying predictors of emergent glucose dysregulation would allow interventions to be directed towards those most at risk, thus creating potential for more precise approaches to clinical and preventative care to reduce the risk of diabetes. In order to have maximum effectiveness, early screening is essential.

Despite these opportunities, further work is needed to clarify how feasible it is to predict T2DM, identify it earlier, or mitigate against its complications in people affected by SMI. Glycated haemoglobin (HbA1c) is a commonly used measure of dysglycaemia and is expected to be regularly recorded in people receiving mental healthcare; for example, it is recommended for English mental health services.[5] Investigating HbA1c recording levels in routine electronic health records (EHRs) is an important first step in this process, including whether this is consistent across all levels of treatment and care; however, to our knowledge this has not received previous attention. First diagnosis in people with psychotic disorders has been identified as the optimal time to screen and prevent physical health conditions developing, including T2DM, but limited data are available on screening levels in this critical time period.[6]

The aim of this study was to investigate rates of HbA1c recording in the electronic mental health records of people with newly diagnosed SMI (schizophrenia, schizoaffective disorder or bipolar disorder presenting for the first time) receiving care from a large mental healthcare provider. Further, we specifically sought to investigate the factors associated with HbA1c levels being recorded or not (including changes in recording rates over time), to determine the extent to which ascertained HbA1c recording could be increased through a novel natural language processing (NLP) algorithm applied to free text, and to describe distributions of HbA1c levels when recorded.

## METHODS
### Sample
The analysed sample was assembled using the Clinical Record Interactive Search (CRIS) data resource at the South London and Maudsley National Health Service (NHS) Foundation Trust (SLAM). SLAM is one of Europe's largest mental healthcare providers, serving over 1.2 million residents of four south London boroughs (Lambeth, Southwark, Lewisham, Croydon). With services covering all ages and specialties, SLAM has fully used EHRs for over 10 years and its CRIS platform, set up in 2008,[7] allows researcher access to de-identified data from the full record within a robust governance framework. Because large volumes of information in mental health clinical records are primarily recorded in text, a range of constructs are now quantifiable at scale using NLP algorithms developed for CRIS.[8]

Using CRIS, we extracted data on all patients with a first diagnosis of contact between 1 January 2008 and 31 December 2018 that resulted in a recorded diagnosis of schizophrenia, schizoaffective disorder or bipolar disorder. As described in the Results, analysed data were subsequently restricted to those from 2010 onwards because of limited data for earlier years. If the patient had more than one HbA1c result, the result closest to diagnosis date was used.

### Measures
Serum HbA1c data were available from two sources: (1) Prestructured imported data from local laboratories from which this assay had been requested by SLAM; (2) Information recorded in free-text fields such as case notes and clinical correspondence. In order to render information available for analysis from the latter, a rules-based NLP algorithm was created to identify HbA1c as a recorded entity and to extract the numerical value attributed to that assay. From a corpus of 250 randomly selected documents, annotation features were first built over a training set. The algorithm rules sought to identify instances of HbA1c recording and the associated score where this was being recorded in relation to the patient (ie, rather than anyone else). Any units (% or mmol/mol) were allowable and any number of decimal places (if any). Primary exclusions were HbA1c mentions without a score—for example, text simply indicating that levels were within the normal range or that a measurement had taken place (ie, without a value described). The algorithm derived from this training set was then evaluated against manually annotated gold standards. Once the precision (positive predictive value) and recall (sensitivity) were judged to be at a good enough standard in the development phase, the algorithm was run over CRIS as a whole using a cloud-based processing platform and the results compiled and evaluated formally. The developed NLP algorithm achieved a precision of 89% and a recall of 93% against the gold standard in an independent manually annotated evaluation data set extracted from CRIS.

The following covariates were additionally extracted: age, gender, ethnicity, first SMI diagnosis recorded (categorised into schizophreniform (including schizophrenia or schizoaffective disorder) or bipolar), and the year of first recorded diagnosis. Recorded glucose level was further evaluated as a measure of dysglycaemia.

### Statistical analysis
The sample was initially described and unadjusted associations were done between first SMI diagnosis with HbA1c recording or not as a binary outcome, first investigating laboratory data alone and then this was supplemented with data from the NLP algorithm. Next, a logistic regression analysis was used to investigate independence of these associations, entering all covariates simultaneously. Finally, given low levels of recording in earlier years, a sensitivity analysis was carried out restricting the sample to those diagnosed in 2010 or after.

### Patient and public involvement
At project planning stage, members of a Service User Advisory Group were shown initial ideas and plans and asked for thoughts and suggestions relating to a broader body of work (a PhD programme for NB) of which this study was a component. The group consisted of services users (of SLAM services) with lived experience of mental health problems. They were very keen on finding out the results of the HbA1c recording and encouraged data linkage to

primary care (which was subsequently carried out for a nested cohort of people with pre-existing diabetes, to be published separately). Suggestions were taken on board and informed, in part, some of the research questions.

## RESULTS

A sample of 21 462 patients was initially characterised based on the study inclusion criteria for the years 2008–2018. From the laboratory data, we ascertained 4106 patients (19.1%) with at least one HbA1c result recorded. When we supplemented this with the NLP algorithm, the number with at least one HbA1c result recorded rose to 6901 (32.2%; a 68% increase); however, the further addition of glucose data did not substantially increase recording results further (table 1). The mean (SD) interval between the date of diagnosis and that of the first recorded HbA1c result was 2.1 (2.5) years.

Factors associated with the recording of HbA1c are displayed in table 1. Considering HbA1c derived from laboratory values or NLP, data were most likely in the 20–29 years age group, in men compared with women, and recording was more common in all minority ethnic groups compared with the white group, particularly so for the black group, and in those with a schizophreniform

**Table 1** Sample characteristics and prevalence of data on HbA1c and/or glucose levels

| | Number | HbA1c—lab value | | HbA1c—lab value or NLP-derived | | HbA1c or glucose | |
|---|---|---|---|---|---|---|---|
| | | % present | $\chi^2$ (df), P value | % present | $\chi^2$ (df), P value | % present | $\chi^2$ (df), P value |
| Total sample | 21 462 | 19.1 | | 32.2 | | 35.2 | |
| **Age group, years** | | | | | | | |
| <20 | 2660 | 20.0 | | 30.3 | | 35.9 | |
| 20–29 | 5237 | 26.0 | 314.9 (4) | 37.9 | 129.9 (4) | 40.6 | 143.36 (4) |
| 30–39 | 4791 | 20.1 | <0.001 | 32.4 | <0.001 | 36.0 | <0.001 |
| 40–49 | 3712 | 16.2 | | 30.9 | | 33.2 | |
| 50+ | 5062 | 12.8 | | 27.8 | | 29.7 | |
| **Gender** | | | | | | | |
| Female | 10 516 | 17.9 | 20.5 (1) | 30.1 | 40.8 (1) | 33.2 | 32.5 (1) |
| Male | 10 942 | 20.3 | <0.001 | 34.1 | <0.001 | 36.9 | <0.001 |
| **Ethnicity** | | | | | | | |
| White | 10 923 | 15.0 | | 26.8 | | 30.2 | |
| Black | 5138 | 29.5 | 467.0 (4) | 47.3 | 663.8 (4) | 49.8 | 584.4 (4) |
| Asian | 1250 | 22.5 | <0.001 | 36.0 | <0.001 | 38.7 | <0.001 |
| Mixed | 631 | 20.3 | | 33.4 | | 36.9 | |
| Other | 1748 | 19.6 | | 32.4 | | 35.2 | |
| **Diagnosis year** | | | | | | | |
| 2008 | 1853 | 11.4 | | 23.0 | | 24.8 | |
| 2009 | 1859 | 9.8 | | 22.9 | | 24.4 | |
| 2010 | 2067 | 11.2 | 730.3 (10) | 23.7 | 458.5 (10) | 25.2 | 588.0 (10) |
| 2011 | 1985 | 12.5 | <0.001 | 25.3 | <0.001 | 27.9 | <0.001 |
| 2012 | 1961 | 14.3 | | 27.5 | | 33.0 | |
| 2013 | 2000 | 18.9 | | 32.3 | | 37.1 | |
| 2014 | 2010 | 23.4 | | 39.7 | | 43.3 | |
| 2015 | 1990 | 24.4 | | 38.4 | | 41.1 | |
| 2016 | 2056 | 29.3 | | 42.0 | | 45.1 | |
| 2017 | 2129 | 29.7 | | 41.3 | | 43.8 | |
| 2018 | 1552 | 24.8 | | 36.5 | | 39.7 | |
| **Diagnosis** | | | | | | | |
| Schizophrenia/ schizoaffective disorder | 14 916 | 21.7 | 213.2 (1) | 36.6 | 452.1 (1) | 39.5 | 414.7 (1) |
| Bipolar disorder | 6546 | 13.2 | <0.001 | 21.9 | <0.001 | 25.1 | <0.001 |

NLP, natural language processing.

**Table 2** Logistic regression analysis of factors associated with recorded HbA1c presence in the electronic mental health record for years 2008–2018

| Characteristic | Unadjusted model 2008–2018 OR (95% CI) | Mutually adjusted model 2008–2018 OR (95% CI) | Mutually adjusted model 2010–2018 OR (95% CI) |
|---|---|---|---|
| Age (10-year increment) | 0.93 | 0.94 | 0.93 |
| | (0.92 to 0.95) | (0.93 to 0.96) | (0.92 to 0.95) |
| Female gender | 0.83 | 0.91 | 0.87 |
| | (0.78 to 0.88) | (0.84 to 0.97) | (0.81 to 0.93) |
| Ethnicity White | Ref. | Ref. | Ref. |
| Black | 2.45 | 2.2 | 2.16 |
| | (2.29 to 2.62) | (2.04 to 2.36) | (2.00 to 2.34) |
| Asian | 1.54 | 1.43 | 1.52 |
| | (1.36 to 1.74) | (1.26 to 1.62) | (1.33 to 1.75) |
| Mixed | 1.37 | 1.21 | 1.14 |
| | (1.16 to 1.63) | (1.02 to 1.45) | (0.94 to 1.37) |
| Other | 1.31 | 1.19 | 1.23 |
| | (1.17 to 1.46) | (1.06 to 1.33) | 1.10 to 1.39) |
| Bipolar disorder diagnosis | 0.49 | 0.57 | 0.57 |
| | (0.45 to 0.52) | (0.53 to 0.61) | (0.53 to 0.62) |
| Diagnosis year (per 2008–2018 increment) | 1.1 | 1.13 | 1.14 |
| | (1.10 to 1.12) | (1.12 to 1.15) | (1.13 to 1.16) |

rather than bipolar disorder diagnosis. Prevalence of recorded HbA1c rose consistently from 2010 apart from the 2018 year of data extraction. In adjusted analyses (table 2), associations with all covariates remained statistically significant; however, ORs were reduced in strength (towards the null) for gender, diagnosis and named ethnic minority groups; they were strengthened for diagnosis year. In those with HbA1c data, mean levels are displayed by covariate status in table 3.

## DISCUSSION

In a large cohort of cases with first diagnosis of SMI drawn from EHRs over a 10-year period, we sought to investigate the level of recording of HbA1c over this time period in a large mental health records database. To our knowledge this is the first such investigation, although attention to dysglycaemia in SMI is an increasing priority in mental healthcare.

A particular feature of this study was that we sought to enhance available data in structured fields with what we believe to be a novel algorithm derived using NLP to identify additional values recorded in text. The level of HbA1c initially available in the analysed sample from structured data fields alone (ie, that imported automatically from laboratory reports) was 19.1%. This rose to 32.2% after the NLP algorithm was applied, representing a clinically important 68% increase, backed up by good performance of the algorithm itself in terms of precision

and recall results when compared with a manually annotated gold standard. This indicates wider provision and knowledge of HbA1c data (eg, from broader laboratory networks and presumably via general practitioner communications) than would be readily available for monitoring purposes at a single provider. It therefore underlines the importance of considering text fields in mental healthcare for sources of data, potentially developing and evaluating algorithms for more widespread multisite use. We also evaluated supplementing data further by looking at whether patients had any glucose level recorded, but this produced only marginal further data and was not considered further.

Considering the availability of HbA1c data in the analysed sample, we found that HbA1c recording has increased over time, although was still only present in 42% of those with SMI at its peak in 2016. Although the overall mean interval from SMI diagnosis to first HbA1c recording was relatively short (2 years), data availability will clearly accumulate over time, hence the slightly lower recording levels in 2018 cases closer to the end of the inclusion period.

Our results suggest that recording rates of HbA1c in mental healthcare remain below 50%, despite electronic enhancements such as automatic transfer of local laboratory results and major quality improvement initiatives such as The Commissioning for Quality and Innovation.[5] Similar findings have been reported from other cohorts,

**Table 3** Distributions of first recorded HbA1c levels

| | Number with at least one HbA1c recording | Mean (SD) HbA1c first recorded |
|---|---|---|
| Total sample | 6901 | 38.3 (10.5) |
| Age group, years | | |
| <20 | 807 | 35.8 (6.0) |
| 20–29 | 1985 | 36.2 (7.2) |
| 30–39 | 1554 | 38.1 (9.8) |
| 40–49 | 1147 | 40.2 (10.0) |
| 50+ | 1408 | 43.5 (14.9) |
| Gender | | |
| Female | 3162 | 38.0 (9.1) |
| Male | 3736 | 38.6 (10.8) |
| Missing | | 34 .0 (2.8) |
| Ethnicity | | |
| White | 2924 | 37.6 (9.6) |
| Black | 2428 | 39.1 (11.2) |
| Asian | 450 | 39.5 (10.1) |
| Mixed | 211 | 38.1 (9.3) |
| Other | 566 | 37.6 (7.7) |
| Missing | 322 | 37.5 (7.2) |
| Year of referral | | |
| 2008 | 426 | 39.8 (11.2) |
| 2009 | 426 | 40.0 (12.6) |
| 2010 | 490 | 39.7 (11.7) |
| 2011 | 503 | 39.5 (13.0) |
| 2012 | 539 | 38.0 (10.4) |
| 2013 | 645 | 38.0 (10.5) |
| 2014 | 798 | 37.2 (9.0) |
| 2015 | 765 | 38.0 (9.8) |
| 2016 | 863 | 38.3 (8.9) |
| 2017 | 880 | 37.5 (9.5) |
| 2018 | 566 | 38.5 (7.5) |
| Diagnosis category | | |
| Schizophrenia/ schizoaffective disorder | 5466 | 38.5 (10.4) |
| Bipolar disorder | 1435 | 37.5 (8.7) |

although we were not able to find any study designs directly comparable to our own. For example, Canadian individuals with diabetes and schizophrenia were found to have lower levels of recommended testing than those with diabetes but without schizophrenia;[9] the former group in that study also had higher rates of diabetes-related hospital visits than the latter, and there is clearly a need to clarify whether lower recorded monitoring results in higher observed adverse consequences. The level of HbA1c recording in people with schizophrenia in that study was 36% over a 2-year period, although the findings are not directly comparable because all of the Canadian cohort had diagnosed diabetes, whereas

ours was an unselected sample. An Australian study also reported on diabetes and prediabetes prevalence (14% and 19%, respectively) in psychiatric inpatients who had had no HbA1c recorded in the preceding 3 months;[10] this again was not directly comparable, although does highlight a general need to screen more regularly.

HbA1c recording has become a focus for UK mental health services in recent years, as a result of government targets being put in place. HbA1c has also gradually taken over from blood glucose as a metric for monitoring or screening for diabetes, although fasting blood glucose levels are preferred after newly starting medication or changes in medication, so that idiosyncratic changes in glucose control can be identified. Although there has been an increase in the years following the introduction of The Commissioning for Quality and Innovation (CQUIN) initiative in 2014,[11] there are still a lot of patients without a record of this important test, and it is important to understand why. One possibility is that levels are checked in primary care but are not transcribed into the mental healthcare record. This is something we anticipate investigating further in future, taking advantage of CRIS linkages with primary care records for a proportion of the mental health service's catchment area;[12 13] however, the lack of recording in the mental health record remains problematic given the level of care required in that sector, the importance of diabetes as a comorbidity in SMI, and the concern about weight gain and impaired glucose tolerance as an adverse effect of some psychotropic medications. Alternatively, as suggested by the Canadian study cited earlier, it is also possible that people with SMI may receive suboptimal screening and monitoring of dysglycaemia across both primary and specialist care sectors, which is also clearly a cause for concern.

We sought to investigate demographic and clinical factors associated with the presence of recorded HbA1c, which we believe is a novel initiative, as we were not able to identify comparable previous research. Associations with younger age may reflect a higher perceived need for screening shortly after onset, and latterly new incentivisation (CQUIN) initiatives around screening were focused particularly on early intervention services; however, it is clearly concerning that recorded levels are less common in older patients who will be at higher risk of dysglycaemia. Higher prevalence of recorded values in men, in non-white ethnic groups and in patients with schizophreniform compared with bipolar disorder diagnoses, on the other hand, may reflect a recognition of higher risk status in these groups.

The mean recorded HbA1c level was 38.3 mmol/mol which is within the normal range, and the SD of 12.3 indicates a sizeable overlap in the distribution of levels with those that would be considered as dysglycaemic (≥48 mmol/mol), consistent with the recognised higher risk in SMI.[14] In a US cross-sectional study of 114 patients with schizophrenia or schizoaffective disorder aged 26–65 years the mean level of HbA1c was reported as 6.1% which approximates as 43.2 mmol/mol.[15] This

is higher than our estimate, although may be accounted for by the relatively high mean age (48.3 years) of that cohort. In studies of patients presenting with first diagnosis psychosis, lower mean HbA1c levels of 5% (approximating 31.2 mmol/mol) have been described[16 17] which may reflect the earlier stage of illness captured. In another review of first-diagnosis psychosis, there were relatively few studies identified providing data on HbA1c[18] which makes this study even more key in understanding how HbA1c is being captured within a mental health setting.

Strengths of the study included the large naturalistic samples, the long time period under investigation and the focus on first diagnosis of SMI, which is a key time for intervention. Considering generalisability, although data are derived from one service provider (SLAM), numerous component teams provide the data and there may therefore be within-provider differences in HbA1c recording, which might be more sizeable than between-provider differences; however, it should be borne in mind that SLAM's catchment contains a highly socially and culturally mixed population and urban and semi-urban settings, resulting in a particular focus on care for SMI, so there is a need for further evaluation and data across broader services and catchments. Considering HbA1c measurement, although the NLP algorithm had a high precision and recall, neither was 100% and therefore there may be findings missed. Given the increase in recording over time, which might have introduced bias from more selective recording in earlier years and preferential monitoring of blood glucose rather than HbA1c in earlier years, we carried out sensitivity analyses restricting to post-2010 results, although these did not alter primary findings substantially. As described above, we were only able to investigate HbA1c recording within mental healthcare and so cannot exclude additional data existing in primary care records; furthermore, we could not identify any other study quantifying the level of primary care HbA1c recording in people with SMI. HbA1c assessment in primary care unknown to secondary care is likely to be minimal in the absence of established diabetes; however, empirical data are required to verify this. Information was also limited on potential confounding factors, including comorbidities, other measures of metabolic syndrome and lifestyle factors such as diet, exercise and smoking. Finally, the focus of the study was on patients at the time of their first diagnosis within a specific mental health service, which does not necessarily equate to the timing of the first ever diagnosis.

In conclusion, the overall level of HbA1c recording was low and further research is needed into ways in which both measurement and recording can be more effectively optimised in these clinical populations.

**Contributors** The study was originally conceived by NB, RS, FG and BS. Data were prepared and analysed by NB, GP and DC. The report was written by NB with inputs from all authors. All authors critically revised the manuscript and approved the final version. NB is the guarantor of this work.

**Funding** NB, FG, GP and RS are part-funded by the National Institute for Health Research (NIHR) Biomedical Research Centre at the South London and Maudsley NHS Foundation Trust and King's College London; FG and RS are part-funded by the National Institute for Health Research (NIHR) Applied Research Collaboration South London (NIHR ARC South London) at King's College Hospital NHS Foundation Trust; RS is part-funded by the DATAMIND HDR UK Mental Health Data Hub (MRC grant MR/W014386). FG is also part-funded by the Maudsley Charity. BS holds a NIHR Advanced fellowship (NIHR301206, 2021-2026). The views expressed are those of the authors and not necessarily those of the NIHR or the Department of Health and Social Care.

**Competing interests** RS declares research support received in the last 3 years from Janssen, GSK and Takeda, and royalties from Oxford University Press. FG has received honoraria from, Lundbeck, Otsuka and Sunovion. BS is on the editorial board of *Ageing Research Reviews*, *Mental Health and Physical Activity*, *The Journal of Evidence Based Medicine*, and *The Brazilian Journal of Psychiatry*. BS has received honorarium from a co-edited book on exercise and mental illness, advisory work from ASICS Europe BV & FitXR for unrelated work.

**Patient and public involvement** Patients and/or the public were involved in the design, or conduct, or reporting, or dissemination plans of this research. Refer to the Methods section for further details.

**Patient consent for publication** Not applicable.

**Ethics approval** The CRIS platform is approved as a source of anonymised data for secondary research use by Oxford Research Ethics Committee C (Reference 18/SC/0372).

**Provenance and peer review** Not commissioned; externally peer reviewed.

**Data availability statement** Data are available upon reasonable request. On request, and after appropriate arrangements, the data employed in this study can be viewed within the secure system firewall.

**ORCID iDs**
Nikeysha Bell http://orcid.org/0000-0002-6504-6443
Gayan Perera http://orcid.org/0000-0002-3414-303X
David Chandran http://orcid.org/0000-0002-0123-666X
Brendon Stubbs http://orcid.org/0000-0001-7387-3791
Fiona Gaughran http://orcid.org/0000-0001-7414-5569
Robert Stewart http://orcid.org/0000-0002-4435-6397

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
