## [Reviewer comments · BMJ Open]

ARTICLE DETAILS

TITLE (PROVISIONAL)	HbA1c recording in patients following a first diagnosis of serious mental illness: The South London and Maudsley Biomedical Research Centre Case Register
AUTHORS	Bell, Nikeysa; Perera, Gayan; Chandran, David; Stubbs, B; Gaughran, Fiona; Stewart, Robert

VERSION 1 – REVIEW

REVIEWER	Bower, Peter University of Manchester, NPCRDC
REVIEW RETURNED	13-Dec-2022

GENERAL COMMENTS	Thank you for the opportunity to comment on this paper. It concerns an important clinical issue (presence of HbA1c measurements in a vulnerable group of patients with SMI). They have used a natural language processing algorithm for extracting recorded values in text fields for missing HBA1C values which adds additional value from a methodological point of view. The data on factors associated with recording was interesting, but I felt that the data on factors associated with higher levels was misplaced – the fundamental issue that the paper deals with is the presence and absence of data, and drifting into issues of high scores seemed to me to be a step too far and added little to the core of the paper, especially since the whole point (to me) is that reporting is poor and therefore not really a good basis (yet) for further analysis because of the possibility of bias. The lack of primary care data is a major issue and one they acknowledge, but is there any data available which might allow an estimate of the size of the 'issue'? They suggest that will be the focus of later work, but can they estimate how much that is likely to be from the wider literature? The NLP methods (and their assessment) are described in some detail, but I am not expert in that area, and I wondered if that needed checking by someone with specific expertise? Is the level of detail enough to make a judgement of the quality of the evaluation of such an algorithm? As this is a major part of the added value I did wonder if this needed a longer description. It is quite a long paper and I think there is a case to remove the data on high HbA1c levels (which, based on such a selective sample, feels quite restricted) and Tables 3 and 4 and provide more detail on the NLP so as to encourage others to take up that innovation. The Discussion is well written but quite long and I felt could be trimmed, as the core messages from the paper are quite simple
---

	and the long discussion lost some of the focus. If they lose the data on correlates of high HbA1c scores, that will sharpen the discussion as well. On page 18 they make a comment about the 'between team' and 'between provider' differences which I did not completely understand, and I was not sure of the evidential basis of the comment either. A minor issue, but when they say they used the HBA1C value "closest to the diagnosis date was used", was it always AFTER the diagnosis date? It is nice to see a description of PPI in a 'big data' study, and I think a few more lines about the composition of the panel (and whether it was specific to this study) would be useful. I would also like to hear about the specific changes that they made in response to the service user input. I consulted with a colleague (Dr Rathi Ravindrarajah) for this review as she has specific expertise in this area.
--	---

REVIEWER	Odhaib , Samih A. University of Basrah, Adult Endocrinology
REVIEW RETURNED	21-Dec-2022

GENERAL COMMENTS	The article is well written and cover an important and relevant subject in the medical practise.
--

VERSION 1 – AUTHOR RESPONSE

Reviewer: 1

Dr. Peter Bower, University of Manchester

Comments to the Author:

Thank you for the opportunity to comment on this paper. It concerns an important clinical issue (presence of HbA1c measurements in a vulnerable group of patients with SMI). They have used a natural language processing algorithm for extracting recorded values in text fields for missing HBA1C values which adds additional value from a methodological point of view.

Thank you for your comments

The data on factors associated with recording was interesting, but I felt that the data on factors associated with higher levels was misplaced – the fundamental issue that the paper deals with is the presence and absence of data, and drifting into issues of high scores seemed to me to be a step too far and added little to the core of the paper, especially since the whole point (to me) is that reporting is poor and therefore not really a good basis (yet) for further analysis because of the possibility of bias. We appreciate the issue raised and have taken this on board. In response, we have removed the second focus of the paper related to HbA1c level. We do feel it appropriate to provide information on distributions of HbA1c level for the sample as a whole and for the clinical/demographic subgroups, as we feel that this would be an obvious question for a reader of this manuscript. We have therefore deleted Table 4 (and relevant text) and have removed the statistical analyses from Table 3, so that data presented are purely descriptive. References to further analyses of HbA1c levels have been removed from the text of the Results and Discussion.

The lack of primary care data is a major issue and one they acknowledge, but is there any data available which might allow an estimate of the size of the 'issue'? They suggest that will be the focus of later work, but can they estimate how much that is likely to be from the wider literature?

As indicated in the manuscript (and a key rationale for this study), there is little to no information on levels of routine investigations in this clinical population. Aside from people with pre-existing diabetes, we would not anticipate meaningful numbers of people with an HbA1c assay known to primary care but not known to specialist mental healthcare, since the rationale for the HbA1c assay is the mental disorder. We have not been able to find any report in the published literature to inform further on this issue. We have added text on the limitation in the Discussion, including this consideration; however, we are aware that it can only be a supposition and requires empirical investigation.

The NLP methods (and their assessment) are described in some detail, but I am not expert in that area, and I wondered if that needed checking by someone with specific expertise? Is the level of detail enough to make a judgement of the quality of the evaluation of such an algorithm? As this is a major part of the added value I did wonder if this needed a longer description.

We have added further text on the algorithm, including more detail on inclusion and exclusion considerations for source annotations, which we hope is now sufficient.

It is quite a long paper and I think there is a case to remove the data on high HbA1c levels (which, based on such a selective sample, feels quite restricted) and Tables 3 and 4 and provide more detail on the NLP so as to encourage others to take up that innovation.

Table 4 has been removed and Table 3 has been modified, as described above.

The Discussion is well written but quite long and I felt could be trimmed, as the core messages from the paper are quite simple and the long discussion lost some of the focus. If they lose the data on correlates of high HbA1c scores, that will sharpen the discussion as well.

The discussion has been abbreviated, as described above. We have retained text on HbA1c levels compared to those reported from previous studies of similar populations, as we feel that this is necessary material for contextualisation.

On page 18 they make a comment about the 'between team' and 'between provider' differences which I did not completely understand, and I was not sure of the evidential basis of the comment either.

This is now on page 17 and has been rephrased for clarity. In essence there are numerous individual mental health teams provided by the South London and Maudsley NHS Foundation Trust (the 'provider' here) which we wanted to highlight, as the approach can vary.

A minor issue, but when they say they used the HBA1C value "closest to the diagnosis date was used", was it always AFTER the diagnosis date?

Yes, we can confirm that the value was always after the diagnosis date. This text has been amended for clarity.

It is nice to see a description of PPI in a 'big data' study, and I think a few more lines about the composition of the panel (and whether it was specific to this study) would be useful. I would also like to hear about the specific changes that they made in response to the service user input.

This has been expanded

I consulted with a colleague (Dr Rathi Ravindrarajah) for this review as she has specific expertise in this area

Reviewer: 2

Dr. Samih A. Odhaib , University of Basrah

Comments to the Author:

The article is well written and cover an important and relevant subject in the medical practise

We are grateful for these supportive comments

VERSION 2 – REVIEW

REVIEWER	Bower, Peter University of Manchester, NPCRDC
REVIEW RETURNED	23-May-2023
GENERAL COMMENTS	Their response to the comments meets my concerns and I am happy with the revision.